# Beyond Human Data: Scaling Self-Training for Problem-Solving with Language Models

**Avi Singh\*, John D Co-Reyes\*, Rishabh Agarwal\*,**

**Ankesh Anand, Piyush Patil, Xavier Garcia, Peter J. Liu, James Harrison, Jaehoon Lee, Kelvin Xu, Aaron Parisi, Abhishek Kumar, Alex Alemi, Alex Rizkowsky, Azade Nova, Ben Adlam, Bernd Bohnet, Gamaleldin Elsayed, Hanie Sedghi, Igor Mordatch, Isabelle Simpson, Izzeddin Gur, Jasper Snoek, Jeffrey Pennington, Jiri Hron, Kathleen Kenealy, Kevin Swersky, Kshiteej Mahajan, Laura Culp, Lechao Xiao, Maxwell L Bileschi, Noah Constant, Roman Novak, Rosanne Liu, Tris Warkentin, Yundi Qian, Yamini Bansal, Ethan Dyer, Behnam Neyshabur, Jascha Sohl-Dickstein, Noah Fiedel**

*\*Equal Contribution, All authors are with Google DeepMind.*
*Correspondence to {singhavi, jcoreyes, rishabhagarwal}@google.com.*

**Reviewed on OpenReview:** *https://openreview.net/forum?id=lNAyUngGFK*

## Abstract

Fine-tuning language models (LMs) on human-generated data remains a prevalent practice. However, the performance of such models is often limited by the quantity and diversity of high-quality human data. In this paper, we explore whether we can go beyond human data on tasks where we have access to scalar feedback, for example, on math problems where one can verify correctness. To do so, we investigate a simple self-training method based on expectation-maximization, which we call ReST$^{EM}$, where we (1) generate samples from the model and filter them using binary feedback, (2) fine-tune the model on these samples, and (3) repeat this process a few times. Testing on advanced MATH reasoning and APPS coding benchmarks using PaLM-2 models, we find that ReST$^{EM}$ scales favorably with model size and significantly surpasses fine-tuning only on human data. Overall, our findings suggest self-training with feedback can reduce dependence on human-generated data.

## 1 Introduction

Large Language Models (LLMs) are revolutionizing the landscape of deep learning, showcasing remarkable capabilities in generating human-quality text and tackling diverse language tasks (Google et al., 2023; OpenAI, 2023). While supervised fine-tuning (SFT) on human-collected data further boosts their performance on tasks of interest, acquiring high-quality human data poses a significant bottleneck. This is particularly demanding for complex problem-solving tasks, requiring significant resources and expert knowledge. To address this hurdle, model-generated synthetic data emerges as a promising alternative, offering scalability and cost-effectiveness, provided its quality can be ensured. While LLMs hold the potential to self-evaluate generated data, this paper explores a simpler setting where an external, scalar feedback signal serves as a quality indicator for each generated sample.

To investigate training on model-generated data, we consider a simple yet powerful self-training approach for language models that requires only two capabilities: 1) generating samples from the model and 2) evaluating these samples with a scoring mechanism. This approach shares similarities with Reinforced Self-Training (ReST) proposed by Gulcehre et al. (2023). We make some modifications to ReST (detailed in Section 3), and call our approach *ReST$^{EM}$*. We show that ReST$^{EM}$ can be viewed as applying expectation-maximization for reinforcement learning (Dayan & Hinton, 1997; Peters & Schaal, 2007), which we present formally in Section 3. Specifically, ReST$^{EM}$ alternates between the expectation and maximization steps:

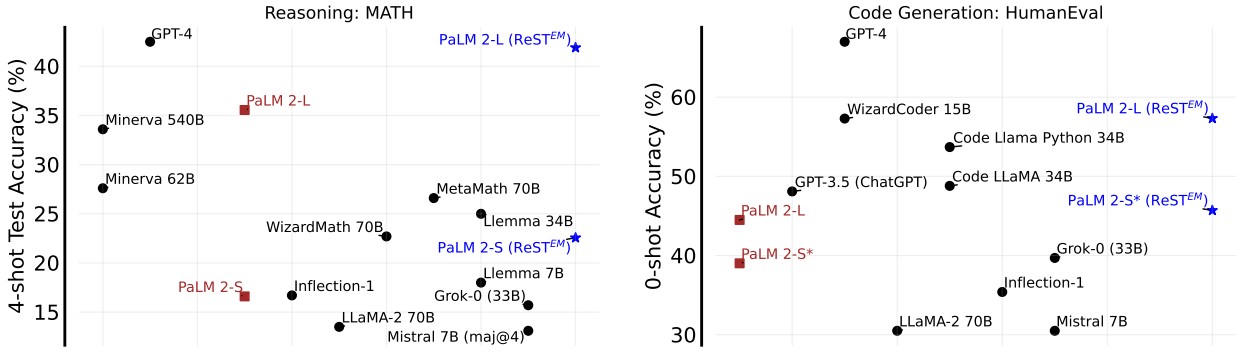

Figure 1: Self-training with ReST$^{EM}$ substantially improves test performance of PaLM 2 models on two challenging benchmarks: MATH and HumanEval. Results for other models are shown for general progress on these tasks and are typically not comparable due to difference in model scales. GPT-4 results are taken from Bubeck et al. (2023). The x-axis approximately denotes release time (not to scale).

1. `Generate` (`E-step`): The language model generates multiple output samples for each input context. Then, we filter these samples using a binary reward to collect the training dataset.

2. `Improve` (`M-step`): The original language model is supervised fine-tuned on the training dataset from the previous `Generate` step. The fine-tuned model is then used in the next `Generate` step.

ReST$^{EM}$, with its various adaptations (Section 4), has demonstrated success in enhancing language models across diverse domains, including machine translation (Norouzi et al., 2016; Gulcehre et al., 2023), semantic parsing (Agarwal et al., 2019), preference alignment (Dong et al., 2023), and elementary reasoning (Zelikman et al., 2022; Yuan et al., 2023). However, prior works primarily applied training with self-generated data to relatively small language models (up to 7B parameters), with limited scalability observed for larger models (Yuan et al., 2023). Complementing these efforts, our work aims to investigate the effectiveness and scalability of model-generated synthetic data compared to human-generated data in two challenging, less explored domains: competition-level mathematical problem-solving (MATH) (Hendrycks et al., 2021b) and code generation (APPS) (Hendrycks et al., 2021a).

Our empirical findings reveal significant advancements in both mathematical reasoning and code generation capabilities when applying ReST$^{EM}$ to PaLM 2 models of varying scales (Figure 1). Notably, models fine-tuned on model-generated synthetic data exhibit remarkably larger performance gains compared to those trained on human-written data (Figure 2, 3). Interestingly, exceeding a couple of iterations of ReST$^{EM}$ leads to diminishing improvement, indicating potential overfitting on small amount of training problems (Figure 4). Additionally, models fine-tuned using ReST$^{EM}$ improve pass@k as well as majority voting performance. Furthermore, these fine-tuned models demonstrate enhanced performance on related but held-out benchmarks, including math problems (GSM8K and Hungarian HS finals), coding (HumanEval), and Big-Bench Hard tasks. We also perform ablation studies to investigate the effect of number of model-generated solutions, training problems, and iterations for ReST$^{EM}$ fine-tuning. Overall, our findings suggest self-training with feedback as a promising approach to reduce dependence on human data.

The key contributions of this work are:

- We introduce ReST$^{EM}$ that enables learning from self-generated data for LLMs, employing a principled expectation-maximization approach within a reinforcement learning framework.

- We demonstrate that training on self-generated solutions surpasses training on human-generated solutions in problem-solving domains, such as mathematics and code generation.

- Through comprehensive ablation studies, we pinpoint the crucial elements necessary for attaining optimal performance.

- LLMs fine-tuned with ReST$^{EM}$ exhibit robust transfer capabilities across various held-out tasks.

## 2 Preliminaries

An autoregressive language model produces an output sequence $\boldsymbol{y} = (y_1, y_2, ....y_T)$ given a context (or source input) $\boldsymbol{x} = (x_1, x_2, ...x_L)$, where the tokens $x_l, y_t$ belong to a fixed vocabulary. Auto-regressive generation involves predicting tokens one at a time, based on the previously generated tokens. Assuming that the model is parameterized by $\theta$, the conditional probability distribution of generating a sequence $\boldsymbol{y}$ given $\boldsymbol{x}$ is

$$p_\theta(\boldsymbol{y} \mid \boldsymbol{x}) = \prod_{t=1}^{T} p_\theta(y_t \mid \boldsymbol{y}_{<t}, \boldsymbol{x}),$$

with the convention $\boldsymbol{y}_{1:0} = \emptyset$ and $\boldsymbol{y}_{1:t-1} = (y_1, y_2, ....y_{t-1})$. For ease of notation, we define $p(y_t|x) := p(y_t|y_{<t}, x)$. The probability of predicting $t^{th}$ token $y_t$, $p(y_t|x)$, is determined using a softmax with temperature $\gamma$: $p(y_t|x) = \frac{\exp(z_t/\gamma)}{\sum_{i=1}^{M} \exp(z_i/\gamma)}$, where $z_t$ is the logit score for the token $y_t$. Higher values of temperature $\gamma$ introduces more randomness, while a lower value makes the output more deterministic by favoring the most probable words.

Given a dataset $\mathcal{D}$ of inputs $\boldsymbol{x}$ and human-generated outputs $\boldsymbol{y}$, supervised fine-tuning (SFT) trains the policy by minimizing the negative log likelihood loss:

$$\mathcal{L}_{\text{SFT}}(\theta) = -\mathbb{E}_{(\boldsymbol{x},\boldsymbol{y})\sim\mathcal{D}} \left[ \sum_{t=1}^{T} \log p_\theta(y_t \mid \boldsymbol{y}_{1:t-1}, \boldsymbol{x}) \right]. \tag{1}$$

We also assume access to a deterministic sequence-level (or terminal) reward $r(\boldsymbol{x}, \boldsymbol{y})$. Then, the reinforcement learning (RL) objective corresponds to:

$$\mathcal{L}_{\text{RL}}(\theta) = \mathbb{E}_{\boldsymbol{x}\sim\mathcal{D}} \left[ \mathbb{E}_{\boldsymbol{y}\sim p_\theta(\boldsymbol{y}|\boldsymbol{x})} \left[ r(\boldsymbol{x}, \boldsymbol{y}) \right] \right].$$

Optimizing $\mathcal{L}_{\text{RL}}$ loss directly using online RL methods, such as policy gradients, requires updating and sampling from the policy numerous times during training. However, the computational cost of fine-tuning on a continual flow of new samples becomes a limitation of online methods, especially when the sizes of the policy network grow to tens or hundreds of billion parameters. We discuss an alternative to such online RL approaches in the next section.

## 3 Expectation-Maximization for Reinforced Self-Training

**Expectation-Maximization (EM) for RL** We first describe the EM-based framework for RL with language models, building upon the prior work by Dayan & Hinton (1997). Let's define a binary optimality variable O, such that $p(O = 1|\boldsymbol{x}, \boldsymbol{y}) \propto f(r(\boldsymbol{x}, \boldsymbol{y}))$, for some non-decreasing non-negative function $f : \mathbb{R} \to \mathbb{R}^+$. We want to maximize the log-likelihood of observing $O = 1$ (obtaining high reward):

$$\log p(O = 1|\boldsymbol{x}) := \log \sum_{\boldsymbol{y}} p_\theta(\boldsymbol{y}|\boldsymbol{x})p(O = 1 \mid \boldsymbol{x}, \boldsymbol{y}).$$

However, the sum over all possible sequences $\boldsymbol{y}$ is typically intractable. Instead of maximizing $\log p(O = 1; \boldsymbol{x})$, one can consider maximizing its ELBO $L(p_\theta, q)$ with respect to parameters $\theta$ and variational distribution $q(y|x)$. Specifically,

$$\begin{aligned}
\log p(O = 1 \mid \boldsymbol{x}) &= \log \mathbb{E}_{q(\boldsymbol{y}|\boldsymbol{x})} \left[ \frac{p(O = 1 \mid \boldsymbol{x}, \boldsymbol{y})p_\theta(\boldsymbol{y} \mid \boldsymbol{x})}{q(\boldsymbol{y} \mid \boldsymbol{x})} \right] \\
&\geq \mathbb{E}_{q(\boldsymbol{y}|\boldsymbol{x})} \left[ \log \frac{p(O = 1 \mid \boldsymbol{x}, \boldsymbol{y})p_\theta(\boldsymbol{y}|\boldsymbol{x})}{q(\boldsymbol{y} \mid \boldsymbol{x})} \right] \quad \text{(Jensen's inequality)} \\
&= \mathbb{E}_{q(\boldsymbol{y}|\boldsymbol{x})} \left[ \log p(O = 1 \mid \boldsymbol{x}, \boldsymbol{y}) \right] - \text{KL} \left[ q(\boldsymbol{y} \mid \boldsymbol{x})||p_\theta(\boldsymbol{y} \mid \boldsymbol{x}) \right] \\
&=: L(p_\theta, q) \tag{2}
\end{aligned}$$

---

**Algorithm 1: ReST (Expectation-Maximization).** Given a initial policy (e.g., pre-trained LM), ReST$^{EM}$ iteratively applies `Generate` and `Improve` steps to update the policy.

---

**Input:** $\mathcal{D}$: Training dataset, $\mathcal{D}_{val}$: Validation dataset, $\mathcal{L}(\boldsymbol{x}, \boldsymbol{y}; \theta)$: loss, $r(\boldsymbol{x}, \boldsymbol{y})$: Non-negative reward
function, $I$: number of iterations, $N$: number of samples per context

**for** $i = 1$ *to* $I$ **do**

    // Generate (E-step)

    Generate dataset $\mathcal{D}_i$ by sampling: $\mathcal{D}_i = \{\ (\boldsymbol{x}^j, \boldsymbol{y}^j)|_{j=1}^N\ \text{s.t.}\ \boldsymbol{x}^j \sim \mathcal{D},\ \boldsymbol{y}^j \sim p_\theta(\boldsymbol{y}|\boldsymbol{x}^j)\ \}$ Annotate $\mathcal{D}_i$
    with the reward $r(\boldsymbol{x}, \boldsymbol{y})$.

    // Improve (M-step)

    **while** *reward improves on* $\mathcal{D}_{val}$ **do**

        Optimise $\theta$ to maximize objective: $J(\theta) = \mathbb{E}_{(\boldsymbol{x}, \boldsymbol{y}) \sim \mathcal{D}_i}\left[r(\boldsymbol{x}, \boldsymbol{y})\ \log p_\theta(\boldsymbol{y}|\boldsymbol{x})\right]$

    **end**

**end**

**Output:** Policy $p_\theta$

---

The EM algorithm (Dempster et al., 1977) for Equation 2 alternates between an E-step and M-step: at iteration $t$, denote the language model parameter to be $\theta^t$ and the variational distribution to be $q^t$.

- **E-step:** $q^{t+1} = \arg\max_q L(p_{\theta^t}, q)$. Since $L(p_{\theta^t}, q)$ can be written as $-KL[q(\boldsymbol{y}|\boldsymbol{x})||q^*(\boldsymbol{y}|\boldsymbol{x})]$, $q^{t+1}(\boldsymbol{y} \mid \boldsymbol{x}) \propto q^*(\boldsymbol{y} \mid \boldsymbol{x}) := p(O = 1|\boldsymbol{x}, \boldsymbol{y})p_{\theta^t}(\boldsymbol{y} \mid \boldsymbol{x})$. Thus, this step is equivalent to weighting the output samples from conditional language model distribution based on their likelihood of obtaining high rewards.

- **M-step:** $\theta^{t+1} := \arg\max_\theta L(p_\theta, q^{t+1}) = \arg\min_\theta KL\left[q^{t+1}(\boldsymbol{y} \mid \boldsymbol{x})||p_\theta(\boldsymbol{y} \mid \boldsymbol{x})\right] = \arg\min_\theta \sum_{\boldsymbol{y}} -q^{t+1}(\boldsymbol{y} \mid \boldsymbol{x}) \log p_\theta(\boldsymbol{y} \mid \boldsymbol{x})$. As such, this step corresponds to maximizing a weighted negative log-likelihood loss.

Alternating between above steps ensures a monotonic improvement in the ELBO: $L(p_{\theta^{t+1}}, q^{t+1}) \geq L(p_{\theta^t}, q^{t+1}) \geq L(p_{\theta^t}, q^t)$.

**EM with non-negative rewards**. If the rewards are non-negative and $f$ is set to the identity function, then $p(O = 1|\boldsymbol{x}, \boldsymbol{y}) \propto r(\boldsymbol{x}, \boldsymbol{y})$ which implies $q^{t+1}(\boldsymbol{y} \mid \boldsymbol{x}) \propto r(\boldsymbol{x}, \boldsymbol{y})p_{\theta^t}(\boldsymbol{y} \mid \boldsymbol{x})$. In this scenario, the updated policy parameters $\theta^{t+1}$ resulting from the M-step at iteration $t$ are given by:

$$\theta^{t+1} := \arg\max_\theta \mathbb{E}_{x \sim \mathcal{D}}\left[\mathbb{E}_{\boldsymbol{y} \sim p_\theta^t(\boldsymbol{y}|\boldsymbol{x})}\left[r(\boldsymbol{x}, \boldsymbol{y}) \log p_\theta(\boldsymbol{y} \mid \boldsymbol{x})\right]\right]. \tag{3}$$

Comparing the above equation with the typical RL objective ($\mathcal{L}_{RL}$) reveals the key distinction between standard RL and EM-based RL: how output data is sampled. Standard RL continuously updates the policy and uses this latest policy to collect data. In contrast, EM-based RL employs a fixed sampling policy from the previous iteration, decoupling data collection from policy optimization. This decoupling in EM-based approaches enables easier scaling to large policy networks, such as LLMs.

**ReST$^{EM}$** Motivated by the EM framework, we now discuss a simplified version of Reinforced Self-Training (ReST) approach by Gulcehre et al. (2023). This approach, which we call ReST$^{EM}$, decouples data collection (E-step) and policy optimization (M-step) in a typical RL pipeline. Algorithm 1 outlines the ReST$^{EM}$ algorithm with multiple iterations, where each iteration corresponds to one `Generate` and `Improve` step. We describe these steps in detail below.

- `Generate` (E-step): In this step, we generate a dataset $\mathcal{D}_i$ by sampling many output sequences from the current policy $p_\theta$: $\mathcal{D}_i = \{\ (\boldsymbol{x}^j, \boldsymbol{y}^j)|_{j=1}^N\ \text{s.t.}\ \boldsymbol{x}^j \sim \mathcal{D},\ \boldsymbol{y}^j \sim p_\theta(\boldsymbol{y}|\boldsymbol{x}^j)\ \}$. Here, the inputs are resampled from the original dataset $\boldsymbol{x}^j \sim \mathcal{D}$. The output sequences in $\mathcal{D}_i$ are then scored with a binary reward function $r(\boldsymbol{x}, \boldsymbol{y})$. In our experiments, we condition the language model using a few-shot prompt with programs for code generation and step-by-step solutions for math problems.

- **Improve** (M-step): In the $i^{th}$ iteration, we use the new dataset $\mathcal{D}_i$ from `Generate` step to fine-tune the policy $p_\theta$. To mitigate task-specific over-fitting, we minimize drift from the base model by always fine tuning the base pretrained language model. For fine-tuning, we minimize the reward-weighted negative log-likelihood loss $J(\theta) = \mathbb{E}_{(\boldsymbol{x},\boldsymbol{y})\sim\mathcal{D}_i}\left[r(\boldsymbol{x},\boldsymbol{y})\,\log p_\theta(\boldsymbol{y}|\boldsymbol{x})\right]$. Once the policy is improved, a new dataset of better quality samples can be created once again.

*Differences with ReST* (Gulcehre et al., 2023). Unlike ReST, we refrain from augmenting $\mathcal{D}_i$ in `Generate` step with human-generated outputs as such data may not always be optimal for learning or it might not be easily available. Furthermore, each `Improve` step fine-tunes the base model instead of the model obtained from the previous ReST iteration. This results in comparable task-specific performance but much better transfer performance on held-out tasks (see Figure 7).

*Remark.* Our experiments focus on problem-solving settings with binary rewards (either 0 or 1), unlike the bounded real-valued rewards assumed by Gulcehre et al. (2023). Specifically, for each `Generate` step, Gulcehre et al. (2023) perform multiple `Improve` steps, where each `Improve` step can be viewed as an M-step with the function $f(r(\boldsymbol{x},\boldsymbol{y})) = r(\boldsymbol{x},\boldsymbol{y}) > \tau$, where $\tau \in \mathbb{R}^+$ increases in successive M-steps. However, with binary rewards, any value of $\tau \in (0,1)$ corresponds to the identical `Improve` steps.

## 4  Related work

Several prior methods can be instantiated using the expectation-maximization framework presented in Section 3. We discuss methods and their relation to ReST$^{EM}$ in this section.

- **Expert Iteration** (ExiT) (Anthony et al., 2017) alternates between two steps: expert improvement and policy distillation. During the expert improvement step (E-step), we combine a base policy with a search procedure to generate samples from a better policy, called the expert policy. Then, in the policy distillation step (M-step), we use these expert samples to train the base policy in a supervised way, effectively improving it to match the expert policy. While ExiT used monte-carlo tree-search, we simply use temperature sampling for collecting samples from the expert policy in ReST. That said, improving the E-step in ReST using the ExIT framework via search and planning procedures with language models would be interesting for future work. For example, Huang et al. (2022) implement a single iteration of ReST$^{EM}$ on simple math reasoning problems. However, unlike our setup, they do not assume access to a correctness reward and instead employ majority-voting (Wang et al., 2023) as a search procedure within the E-step.

- **Self-Taught Reasoner** (STaR) (Zelikman et al., 2022) employed greedy decoding instead of temperature sampling for the E-step in ReST$^{EM}$, which is restricted to one model-generated solution per problem during data collection. Additionally, STaR proposed rationalization as an alternative to temperature sampling, where the language model is provided with the correct answer as part of the input to generate correct solutions for difficult problems. However, in our preliminary experiments, rationalization leads to substantial increase in false positive solutions that result in correct answer but with incorrect reasoning.

- **Rejection Sampling Fine-tuning** (RFT) (Yuan et al., 2023) improves reasoning performance on GSM8K and corresponds to running a single generate (E-step) and improve (M-step) of ReST$^{EM}$. While RFT demonstrated limited performance improvements on GSM8K with increasing language model capacity, ReST$^{EM}$ achieves larger gains on more challenging APPS and MATH benchmarks when scaling PaLM 2 model capacity. Moreover, we observe that using multiple iterations of ReST$^{EM}$ result in larger performance gains.

- **Iterative Maximum Likelihood** (IML) optimizes a policy using a reward-weighted log-likelihood objective on self-collected data. IML has been shown to perform well with relatively small-scale language models for semantic parsing (Liang et al., 2016; Agarwal et al., 2019), machine translation (Wu et al., 2016) and simple math reasoning (Ni et al., 2022). Each E-step and M-step in IML

| | ReST$^{EM}$ | ReST | STaR | RFT |
|---|---|---|---|---|
| Starts from fine-tuned model | ✗ | ✓ | ✗ | ✗ |
| Finetunes from base model in each iteration | ✓ | ✗ | ✓ | N/A |
| Uses rationalizations for unsolved questions | ✗ | ✗ | ✓ | ✗ |
| Temperature sampling for exploration | ✓ | ✓ | ✗ | ✓ |
| Experiments with Large LMs | ✓ | ✗ | ✗ | ✓ |
| Multiple iterations | ✓ | ✓ | ✓ | ✗ |
| Larger gains on bigger models | ✓ | N/A | N/A | ✗ |
| Evaluation on held out tasks | ✓ | ✗ | ✗ | ✗ |

Table 1: Differences between ReST$^{EM}$ and other closely related approaches utilizing synthetic data for advancing language model capabilities.

is performed over a mini-batch of training examples instead of the entire training dataset, as done in ReST$^{EM}$. In IML, the learned policy can significantly diverge from the initial pretrained model, which can manifest as task-specific overfitting, where the model performs well on the target task but loses its ability to generalize to other tasks or domains. Additionally, the tightly coupled nature of data collection and policy optimization in IML leads to high computational cost with large LMs, making it significantly more expensive than ReST$^{EM}$.

- **Reward weighted regression** (RWR) (Peters & Schaal, 2007) corresponds to EM where we set $p(O = 1|\boldsymbol{x}, \boldsymbol{y}) \propto \exp\left(r(\boldsymbol{x}, \boldsymbol{y})\right)$ in Section 3. RWR has been previously applied to robotic control, as it can be easily applied to non-binary reward functions. Norouzi et al. (2016) build on RWR to propose a general variant of IML for machine translation.

- **Reward ranked fine-tuning** (RAFT) (Dong et al., 2023) can be interpreted as alternating between E-step and M-step over mini-batches, where E-step uses the the output sample with maximum reward for each input context. For binary reward functions, RAFT is analogous to IML and as such, can be viewed as an instantiation of ReST$^{EM}$.

**Other related works**: TRICE (Phan et al., 2023) proposes an EM-based approach to maximize the marginal log-likelihood (MML) of generating a correct answer for a reasoning problem, where the chain-of-thought rationale is treated as a latent variable. While E-step in ReST$^{EM}$ simply corresponds to sampling from the model and filtering with a binary reward, TRICE uses Markov-chain Monte Carlo with a control variate to approximate the MML gradient. Sordoni et al. (2023) propose a gradient-free EM-based approach, similar to RAFT, for prompt-optimization for frozen LLMs.

## 5 Experiments and analysis

The goal of our experiments is to answer the following questions:

1. How effective is ReST$^{EM}$ compared to fine-tuning on human-generated data?

2. How many iterations are needed for optimal performance? How quickly does ReST$^{EM}$ leads to overfitting on training set?

3. How does ReST$^{EM}$ affect pass@k and majority voting performance?

4. If we fine-tune using model-generated data on a specific task, do we see positive transfer to related tasks? Is there any performance degradation compared to the base model when evaluating our fine-tuned models on a broad suite of tasks?

5. How much input data do we need to get most of the performance gains from ReST$^{EM}$? Is one iteration of ReST$^{EM}$ sufficient?

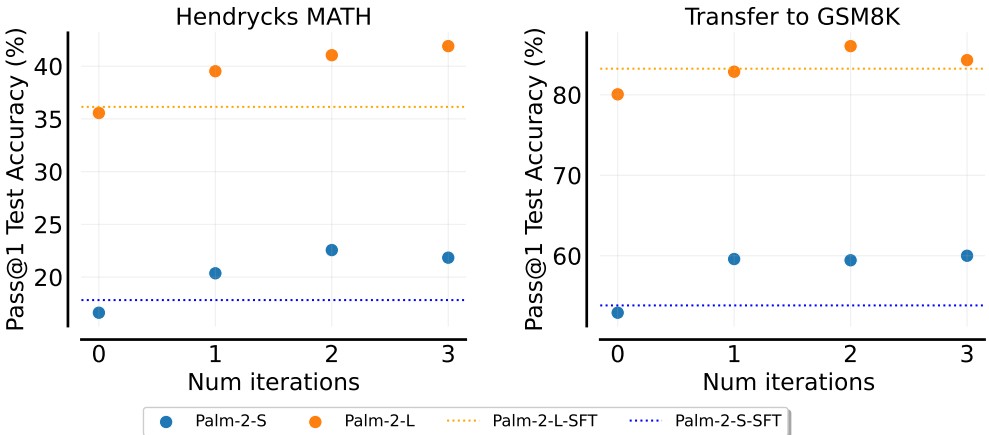

Figure 2: **ReST$^{EM}$ for math problem-solving**. Test performance on MATH and GSM8K (transfer) for PaLM 2-S* and PaLM 2-L as a function of ReST$^{EM}$ iterations. We also report performance of models fine-tuned via SFT on human-generated data as a baseline. Iteration 0 corresponds to pre-trained model performance. Following Google et al. (2023), we use greedy decoding for evaluation.

**Training Datasets**. We evaluate ReST$^{EM}$ primarily on mathematical problem solving using the Hendrycks' MATH dataset (Hendrycks et al., 2021b) and code generation using the APPS (Introductory) dataset (Hendrycks et al., 2021a). MATH and APPS (Introductory) contain 7500 and 2342 training problems respectively. We select these tasks because the model outputs can be automatically evaluated as correct or incorrect, perfectly suited for ReST$^{EM}$. Both these datasets offer binary rewards: on MATH, model-generated answers can be easily verified for correctness using the ground-truth answer, while on APPS, test cases determine whether the generated code is correct.

**Models**. We use the PaLM 2 models (Google et al., 2023) with public APIs on Google Cloud for experiments, including PaLM 2-S (Bison), PaLM 2-S* (Codey), and PaLM 2-L (Unicorn).

**Evaluation**. We report generalization performance using the test splits of the MATH and APPS (Introductory) datasets. For measuring transfer performance, we look at GSM8K (Cobbe et al., 2021), Hungarian HS finals (Paster, 2023), and HumanEval (Chen et al., 2021) datasets. We also evaluate our models using the Big-Bench Hard (Suzgun et al., 2022) benchmark to evaluate general capabilities. All evaluations follow the settings from Google et al. (2023), unless specified otherwise.

**Implementation Details**. During each iteration of ReST$^{EM}$, we generated a fixed number of solutions per problem for the E-step: 32 for the MATH dataset and 64 for the APPS dataset. For generating solutions, we sample from the language model using top-K sampling with K=40 and temperature of 0.7. However, directly using all these model-generated solutions can lead to an imbalanced dataset, as we will have a lot more correct solutions for the easier problems. To mitigate this, we introduced a cut-off threshold for the maximum number of solutions per problem, a design choice also used by Zelikman et al. (2022), included in the fine-tuning dataset: 10 for both MATH and APPS. This approach ensures diversity in the training data and safeguards against overfitting on easier problems. For fine-tuning, we use the few-shot prompt (and the question) as input to the model, and use the model-generated solutions as targets. We only apply the next token prediction loss (Equation 1) on the targets. Due to the cost of our experiments (thousands of TPU hours for every fine-tuning run), each experiment is performed once.

## 5.1 ReST$^{EM}$ on MATH and APPS

Figures 2 and 3 show the performance of ReST$^{EM}$ when trained on the MATH and APPS datasets, respectively. We see that MATH benefits from performing multiple iterations of ReST$^{EM}$, both in terms of performance on the MATH test set, as well as transfer to GSM8K. On the other hand, we see that most of

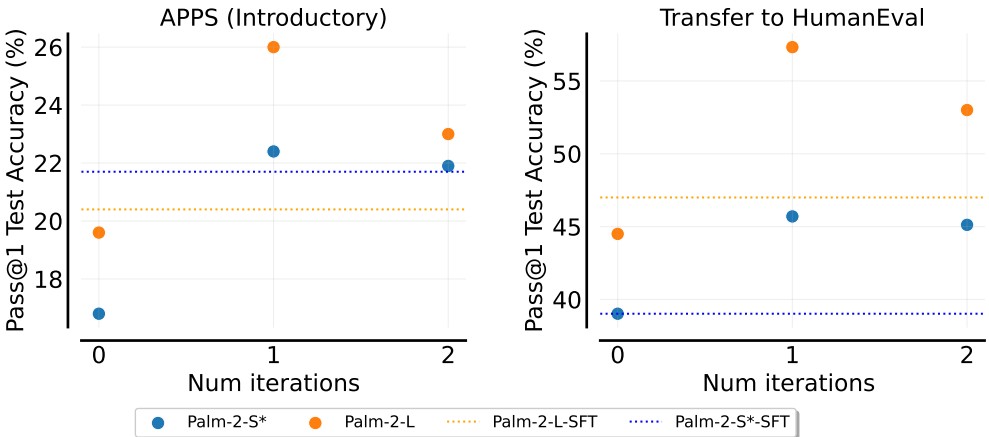

Figure 3: **ReST$^{EM}$ for code-generation**. Test performance on APPS (introductory) and HumanEval (transfer) for PaLM 2-S* and PaLM 2-L as a function of ReST$^{EM}$ iterations.

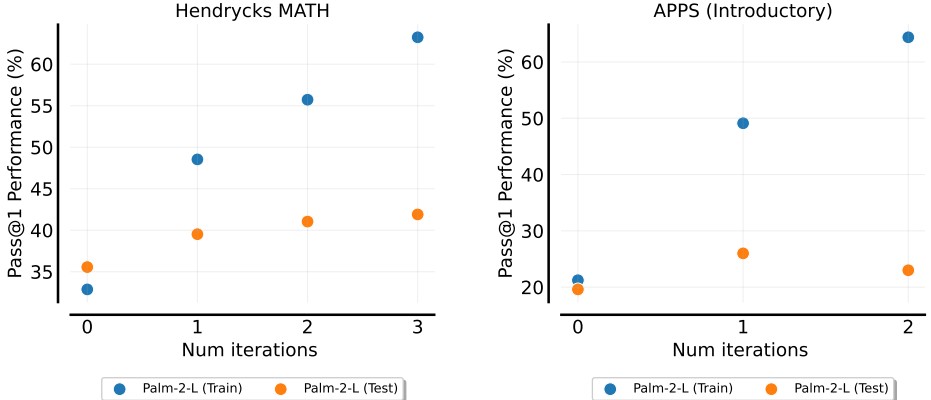

Figure 4: **Train-test performance gap** on (left) MATH with PaLM-2-L, and (right) APPS with PaLM-2-S*, as a function of ReST$^{EM}$ iterations.

the gains for APPS come from the first iteration, and the performing more iterations leads to a regression in performance on both APPS and HumanEval.

Interestingly, Figures 2 and 3 demonstrate that fine-tuning on model-generated solutions substantially outperforms using human-written solutions, especially for the PaLM 2-L model. This aligns with findings of Yuan et al. (2023) and recent work on distilling LLMs using model-generated data (Agarwal et al., 2023; Gu et al., 2023). However, unlike Yuan et al. (2023), who observed diminishing returns from model-generated data on GSM8K when scaling model capacity, our results suggest an opposite trend: ReST$^{EM}$ leads to larger performance gains as model capacity increases. On the MATH dataset, the test accuracy improvement with ReST$^{EM}$ is 5.94% for PaLM 2-S compared to 6.34% for the larger PaLM 2-L model. Similarly, on the APPS dataset, improvements are 5.6% for PaLM 2-S* compared to 6.4% for PaLM 2-L. This is in addition to the fact that the larger models start with a much stronger initial performance, and improvements on these benchmarks generally get harder as the baseline performance goes up.

**Train-test performance gap**. Figure 4 shows that while training performance increases linearly with the number of ReST$^{EM}$ iterations, test set performance does not. For MATH, test performance improvements are small after the first iteration, and for APPS, we observe a regression in performance in the $2^{nd}$ iteration. We suspect that the regression in performance is likely due to overfitting on the small set of training problems. Since the APPS dataset is about a third of the size of the MATH dataset, it suffers more from this problem.

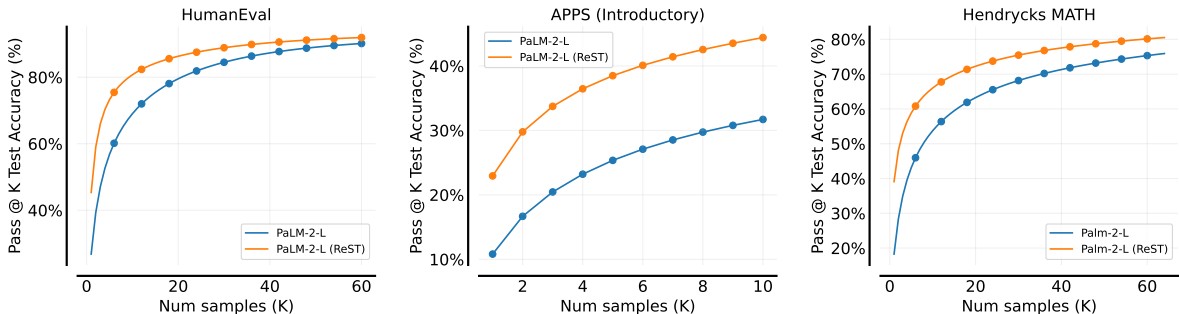

Figure 5: **Pass@K results** for PaLM-2-L pretrained model as well as model fine-tuned with ReST$^{EM}$. For a fixed number of samples K, fine-tuning with ReST$^{EM}$ substantially improves Pass@K performance. We set temperature to 1.0 and use nucleus sampling with $p = 0.95$.

## 5.2 Impact on Pass@K and Majority-Voting Performance

To investigate the impact of fine-tuning with ReST$^{EM}$ on the diversity of the final model's generated outputs, we evaluate pass@k (Chen et al., 2021) and majority voting (Wang et al., 2023) performance of the fine-tuned PaLM 2-L model relative to the base model.

**Pass@K** measures the probability that at least one of the K generated solutions for a problem is correct, that is, outputs the correct answer for math problems or passes all the unit tests for code generation. Figure 5 shows the performance of Palm-2-L on the pass@K metric. We see that model obtained after ReST$^{EM}$ fine-tuning is stronger for all values of K, with the performance gap typically being the highest for K=1.

**Majority voting** first samples a diverse set of reasoning paths instead of only taking the greedy one, and then selects the most consistent answer by marginalizing out the sampled reasoning paths. For Hendrycks MATH, it is possible to use majority voting to maximize Pass@1 performance, and we find that when using 64 samples per question, the PaLM 2-L fine-tuned with ReST$^{EM}$ obtains a test accuracy of **48.82**, while the base model gets 44.02.

## 5.3 Ablation Studies

**Impact of multiple iterations** Our results show that multiple iterations can sometimes lead to overfitting on the train set (Figure 4). This raises the question of whether multiple iterations are really necessary. Is it better to collect a larger dataset and perform just a single iteration of ReST$^{EM}$? To investigate this, we collect a dataset with the base PaLM-2-L model on Hendrycks MATH that is 3× as many solutions per problem as used in a single iteration of ReST$^{EM}$ for the E-step. Fine-tuning with this dataset results in pass@1 performance of 40.3%, which is lower than the 41% in second and 41.9% in third iteration, as shown in Figure 2. These results indicate that performing multiple iterations of ReST$^{EM}$ leads to higher performance compared a single iteration with 3x the data.

**Comparing model-generated data with human data** A key strength of ReST$^{EM}$ is its ability to generate multiple correct solutions for each problem. This provides valuable additional training data compared to human-generated data, which typically offers only a single solution per problem. While this makes a comparison in Figures 2 and 3 not entirely fair, it also highlights the potential of ReST$^{EM}$ to boost performance with diverse and correct solutions.

In order to enable an apples-to-apples comparison, we conduct the following study: we select all Hendrycks MATH questions for which we have at least one correct model-generated solution, resulting in about 5K questions. For these 5K questions, we run two fine-tuning experiments: SFT(5K) where we fine-tune on human-written solutions (one per question), and ReST$^{*}$(5K) where we fine-tune on model-generated solutions (also one per question, selected at random).

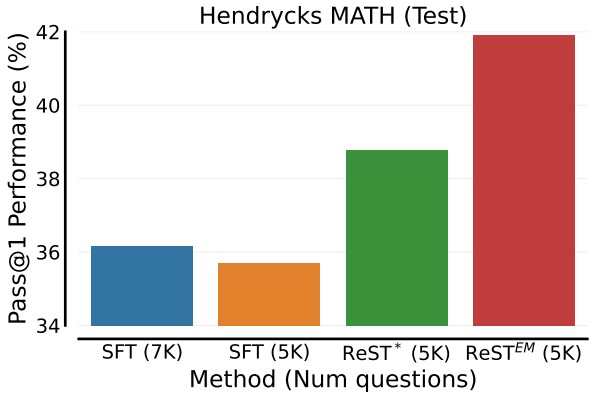 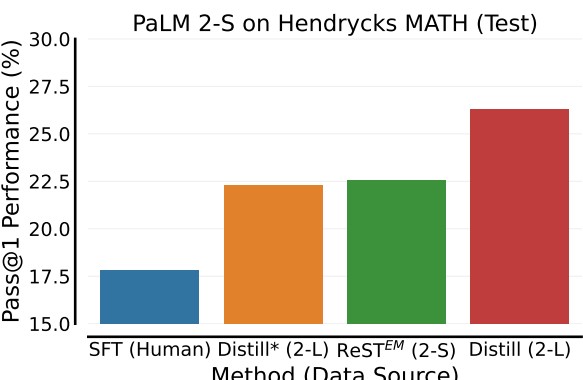

Figure 6: **Left**. Comparing ReST$^{EM}$ with SFT on MATH. SFT refers to fine-tuning on human data, while ReST* refers to a version of ReST$^{EM}$ with one iteration that uses only one correct sample per problem. Here, ReST denotes ReST$^{EM}$ with 3 iterations. For each method, we denote the number of questions in parenthesis. **Right**. Impact of Model-Generated Data for Distillation.

The results in Figure 6 (right), show that ReST$^{EM}$ outperforms fine-tuning on human data even in this much more restricted setting. Furthermore, the efficacy of ReST(5K) over ReST*(5K) highlights the additional gain in performance that we can obtain by spending more compute on sampling a large number of solutions and performing multiple iterations of ReST$^{EM}$.

**Distillation with ReST$^{EM}$-generated data** The above results indicate that self-generated data can be better than human data for fine-tuning language models. We hypothesize this may be because model-generated solutions are more in-distribution compared to human-written solutions. This raises the question of whether ReST$^{EM}$-generated data can benefit different models than the one generating the data.

To answer this question, we consider a distillation setup on MATH where we fine-tune PaLM 2-S using data generated by PaLM 2-L, resulting in solutions for about 5K questions. Specifically, we ran two distillation experiments: Distill* (2-L) where we fine-tune on teacher-generated solutions (one per question), similar to ReST (5K), and Distill (2-L), which includes multiple solutions per problem, generated during the final iteration of ReST$^{EM}$ with PaLM 2-L.

Our results, shown in Figure 6 (right), reveal that Distill* surpasses the performance achieved by fine-tuning on human-written solutions, despite having smaller number of training questions. Additionally, fine-tuning PaLM 2-S with multiple solutions from PaLM 2-L, namely Distill (2-L), is superior than using self-generated solutions via ReST$^{EM}$. This improvement is likely due to the larger number of training questions with solutions in PaLM 2-L generated data compared to 2-S. Overall, these results indicate that model-generated data can be more effective for fine-tuning smaller models than relying on human-generated data.

**ReST *vs* ReST$^{EM}$** One of the main differences between ReST and ReST$^{EM}$ is that ReST$^{EM}$ always fine-tunes the base model for each iteration while ReST continues to finetune the the model from the last iteration. We run an ablation comparing these options using PaLM 2-S* in Figure 7 and observe that while ReST and ReST$^{EM}$ have similar performance on APPS, the transfer performance to HumanEval is substantially better with ReST$^{EM}$.

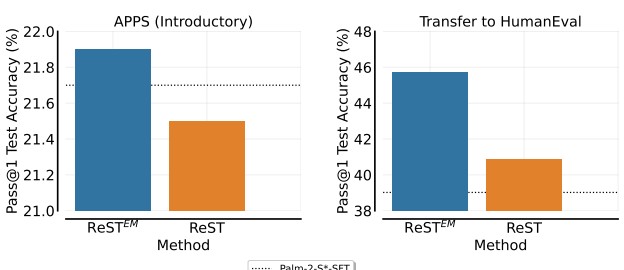

Figure 7: ReST$^{EM}$ *vs* ReST using PaLM 2-S*.

**Impact of dataset size** Since one of the main ingredients needed for ReST$^{EM}$ is a dataset of input contexts (e.g., questions for MATH), we are interested in evaluating the effect of number of input problems.

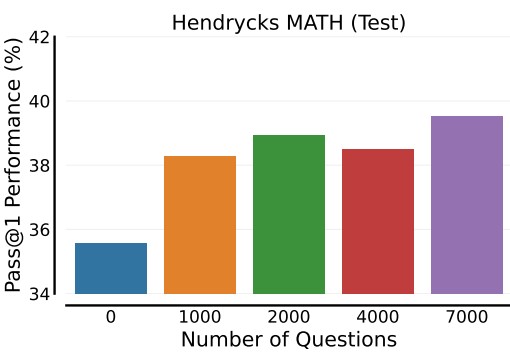 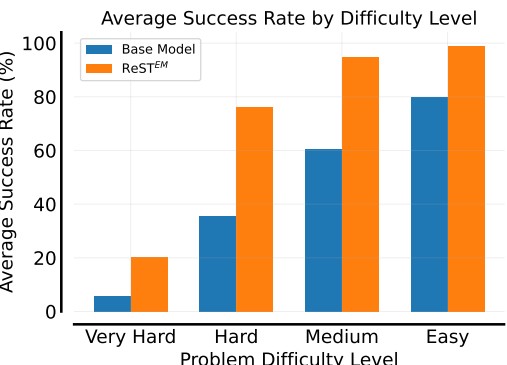

Figure 8: **Left**. Performance for a *single iteration* of ReST$^{EM}$ as a function of dataset size (number of questions) on MATH. **Right**. Improvement from ReST$^{EM}$ based on the difficulty level of the question.

The results from our dataset ablations using the PaLM-2-L model on Hendrycks MATH, Figure 8 (left), show that utilizing just 1000 MATH questions results in significant gains, implying that the method is very efficient in the number of prompts needed. However, we noted a slight decrease in performance when using 4,000 questions compared to 2,000, indicating potential variance in the fine-tuning process. Ideally, conducting this experiment multiple times would help quantify this variance, but this is prohibitively resource-intensive. Overall, we find that ReST$^{EM}$ is quite sample efficient and performance gains from ReST$^{EM}$ improve as we increase the dataset size.

**Which Questions Benefit Most from ReST$^{EM}$** We evaluate the performance enhancement of ReST$^{EM}$ across different question difficulties in the Hendrycks MATH dataset. Questions are classified based on success rates from the base model at a temperature setting of T=1.0 into four categories: "easy" (answered correctly 75%-100% of the time), "medium" (50%-75%), "hard" (25%-50%), and "very hard" (below 25%). Figure 8 (right) presents the average success rates for these categories, comparing the base model to the ReST$^{EM}$-finetuned model. The results demonstrate that ReST$^{EM}$ consistently improves performance across all difficulties, with the highest gains coming for questions categorized as medium and hard.

### 5.4 Impact on Reasoning capabilities

**General capabilities**. BIG-Bench provides a suite of over 200 tasks that can be used to probe LLMs' performance across a range of fields and capabilities. BIG-Bench Hard (BBH) (Suzgun et al., 2022) is a subset of 23 BIG-Bench tasks where the previous generation of LLMs, such as Codex and PaLM 540B, performed below the average human rater. We follow the protocol of Google et al. (2023) and evaluate on BBH using both few-shot and chain-of-thought prompting. Figure 9 shows the performance of ReST$^{EM}$-finetuned models, and compares them against the base PaLM-2 model. We see no major degradation on any of the BBH tasks. Furthermore, the model fine-tuned on Hendrycks MATH outperforms the base model on this suite when using chain-of-thought prompting, and the model fine-tuned on APPS also shows slight performance gains. When using direct prompting, all three models perform similarly.

**Problem-solving**. To stress test the math problem-solving capabilities on a held-out "real-world" evaluation set, we evaluate our model on the 2023 Hungarian high school finals exam in mathematics, akin to Grok. We follow the evaluation protocol from Paster (2023). Specifically, we evaluate the PaLM 2-L model, fine-tuned with ReST$^{EM}$ on Hendrycks MATH, using the 1-shot prompt from Grok, sample solutions using temperature 0.1, and manually grade the outputs using the rubric provided by the examiners. The results from evaluation are shown in Figure 10. We find that PaLM-2-L fine-tuned with ReST$^{EM}$ performs well on this exam, surpassing the performance of all existing models except GPT-4.

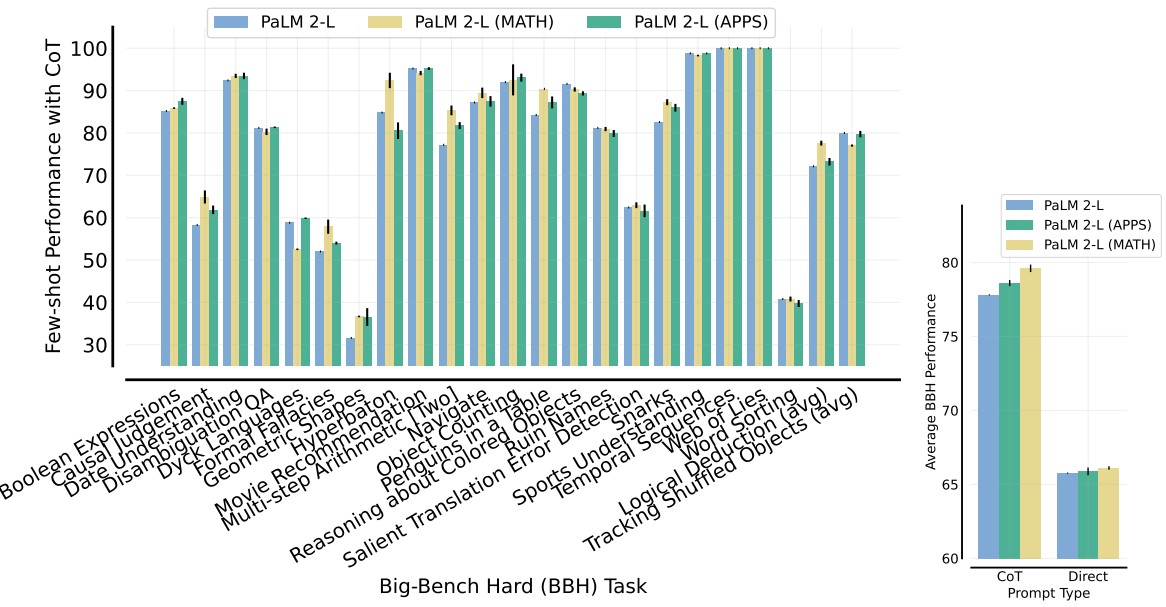

Figure 9: Comparing the ReST$^{EM}$ models to the base model on the Big-Bench Hard suite of tasks. Evaluations were conducted across multiple checkpoints, and the vertical black lines denote standard deviation.

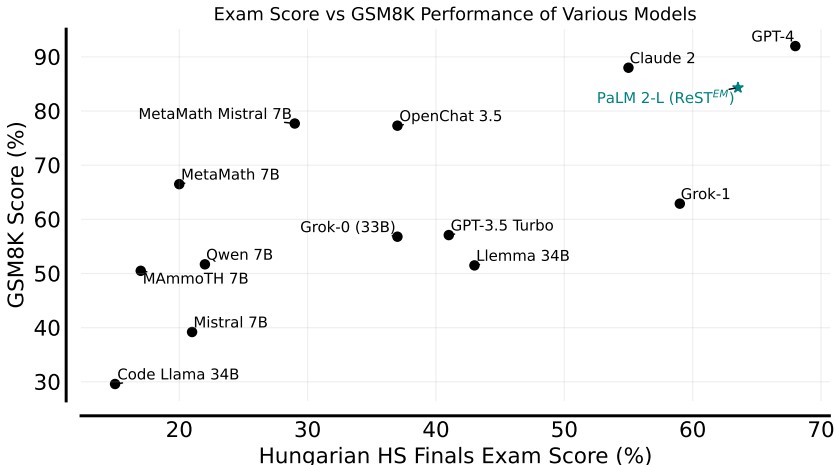

Figure 10: **Transfer results on Hungarian HS Finals Exam.** Results for models other than PaLM-2-L finetuned with ReST$^{EM}$ are taken from Paster (2023). Several models specialized for mathematics perform well on the widely-used GSM8K benchmark but perform poorly on the Hungarian exam. In contrast, PaLM 2-L model fine-tuned with ReST$^{EM}$ performs well on both these benchmarks.

## 6 Discussion

In this paper, we propose training on model-generated data combined with a reward function, via ReST$^{EM}$, for improving the performance of LLMs on problem-solving tasks. Furthermore, we demonstrate that ReST$^{EM}$ is theoretically grounded in the application of expectation-maximization to RL. We evaluate ReST$^{EM}$ on mathematical problem solving and code generation, and show that ReST$^{EM}$ offers significant performance gains at a relatively low computational cost, especially when compared to the cost of pre-training. Our experiments also show that ReST$^{EM}$ does not lead to regression on other tasks. We conduct a number of ablations to better understand the strengths and weaknesses of this method, and find that it is data-efficient, but also requires some vigilance to avoid over-fitting.

There are a number of limitations associated with ReST$^{EM}$. First, this method requires a moderately-sized training set of problems or prompts, which would need to be collected (from humans) for any new task of interest. Second, ReST$^{EM}$ also requires access to a manually-designed or learned reward function, ideally one that can be computed automatically. Finally, while ReST$^{EM}$ allows significant performance improvements in pass@1 performance, it may not quite close the gap to pass@K performance for the same task (with a sufficiently large K). Future research in self-improvement in language models should focus on automating manual parts of the pipeline (likely through language models as well), and explore algorithmic improvements that reduce the gap to pass@K performance.

## Acknowledgements

We would like to thank Tom Le Paine for providing feedback to an early draft. We also acknowledge Benjamin Anderson, Sridhar Thiagarajan, Feryal Behbahani, Aleksandra Faust, Doina Precup, Olivier Bachem, and Slav Petrov for helpful discussions.

## Author Contributions

Avi, Rishabh, and JD jointly led the project. Avi was responsible for training and evaluation infrastructure, ablations and experiments on MATH, JD led the experiments on APPS, Rishabh was responsible for the paper writing, evaluations, and distillation ablations.

Ankesh, Piyush, Ethan, and Behnam observed preliminary findings about efficacy of model-generated data on MATH for Minerva models and motivated this research. Piyush also helped Avi in setting up infrastructure. Xavier, Peter, James, Jaeheoon, Kelvin and Yamini took part in project discussions. Jascha and Noah sponsored and advised the project. All other authors provided feedback on this work.

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
