# OpenReview forum: "Beyond Human Data: Scaling Self-Training for Problem-Solving with Language Models"
_TMLR — Accepted by TMLR_

### Review · Reviewer_Un2a · 2024-02-10

**Summary Of Contributions:**

The authors describe a framework, $ReST^{EM}$, to iteratively improve language models on tasks for which there is a binary correctness reward. At each round, (1) samples are generated from the model, and (2) the model is fine-tuned on the correct samples. The authors present how the approach is related to expectation-maximization. Experiments are run primarily on math and coding tasks, using PaLM models. In addition to the final performance, the paper also examines generalization to other datasets, distillation, etc.

**Audience:**

Yes

**Claims And Evidence:**

Yes

**Requested Changes:**

**Critical**

- What is the "x" axis in figure 1? Please update the figure.

- List your contributions more clearly in the introduction. Before getting to the related work section, it was difficult to understand what is proposed here and what already exists in the literature. The paragraph starting with "$ReST^{EM}$, with its various adaptations (Section 4)" should likely be reworked.

- In figure 3 (left side), why is Palm-2-S*-SFT better than Palm-2-L-SFT? Is that an error or is it correct?

**Would strengthen the work**

I would be interested in seeing how the model improves based on the initial difficulty of a question. You could bucket questions based on the initial success rate (out of the 40 samples), then show the performance within each bucket at different iterations (or maybe just the final results). Are there some questions with 0% initial success rate that can be answered later on?

**Minor**

In Eq. 1, it looks like T is a constant, but it depends on each target y. You could add a subscript to make that clearer.

**Other**

- Why is the approach named based on ReST instead of rejection-sampling fine-tuning or other similar methods?

- On page 3, for the E-step, should there be a negative sign in front of $KL[q(y|x)||q^*(y|x)]$? Also, there seems to be an additional vertical bar in that expression that shouldn't be there.

- Could you briefly expand the derivation of the equation in the M-step? [First in the response, then possibly in the paper]

**Strengths And Weaknesses:**

**Strengths**

- The method is simple, yet can be more effective than standard fine-tuning.

- The experiments are pretty comprehensive (although see weaknesses and requested changes for some suggestions). They generally satisfactorily answer the questions posed by the authors at the beginning of the experimental section.

- The approach is justified theoretically [although I still need to take a closer look at some details].

**Weaknesses**

- Reinforced Self-Training (Gulcehre et al., 2023) should likely be described in more detail. If a reader is not familiar with it, it can be difficult to understand exactly what it does, and how $ReST^{EM}$ relates to it.

- Given that many practitioners already fine-tune models on human-generated data, they would possibly also be interested in the results if you apply $ReST^{EM}$ on an already fine-tuned model, instead of just the pre-trained one.

- I found section 3 (Expectation-Maximization for Reinforced Self-Training) slightly difficult to follow.

---

> ### Author Response · Authors · 2024-02-23
> **Response to reviewer**
>
> “What is the "x" axis in figure 1? Please update the figure.”
> - We originally made the x-axis in Figure 1 to be time. However, this resulted in an illegible figure since a lot of the models had come out very recently (at the time of compilation). X-axis still approximately denotes time, but it is not to scale, and there are some slight changes to make the figure readable. We have updated the figure caption with this information.
>
> “List your contributions more clearly in the introduction.”
> - We have added a new contribution section in the introduction.
>
> “Why is Palm-2-S*-SFT better than Palm-2-L-SFT?”
> - Palm-2-S*-SFT is indeed better than Palm-2-L-SFT on APPS. This is not too surprising because Palm-2-S* is finetuned specifically for code. We do see that Palm-2-L has better transfer to HumanEval than Palm-2-S*. In Figure 7 (left), we see a similar pattern where ReST, which does continual finetuning of a model on APPS, does worse on transfer to HumanEval compared to ReST$^{EM}$.
>
> “I would be interested in seeing how the model improves based on the initial difficulty of a question”
> - We have added an analysis of which questions benefit the most from ReST$^{EM}$ in Figure 8. As suggested by the reviewer, questions are classified based on success rates from the base model at a temperature setting of T=1.0 into four categories: “easy” (answered correctly 75%-100% of the time), “medium” (50%-75%), “hard” (25%-50%), and “very hard” (below 25%). The average success rates for these four buckets demonstrate that ReST$^{EM}$ improves performance across all difficulties, with the highest (absolute) gains coming for questions categorized as medium and hard.
>
> "Why is the approach named based on ReST instead of rejection-sampling fine-tuning or other similar methods?"
> - ReST seemed a more natural framework than RFT as RFT is a single iteration method (exactly E-step and M-step), while our proposed ReST$^{EM}$ is naturally multi-iteration. As such, RFT can be seen as a special case of ReST. Similarly, we do not build on STaR, as STaR uses rationalization to include answers in prompt (which leaks test-cases for code generation and results in false positives on MATH) as well as used greedy decoding for E-step, restricting to 1 solution per problem for data collection.
>
> "On page 3, for the E-step, should there be a negative sign in front of $KL[q(y|x)||q^*(y|x)]$? Also, there seems to be an additional vertical bar in that expression that shouldn't be there."
> - Indeed, the negative sign was missing. We fixed the typo as well.
>
> "Could you briefly expand the derivation of the equation in the M-step? [First in the response, then possibly in the paper]"
> - The first term $E_q[log p(O=1|x,y)]$ is independent of \theta, so the arg max_\theta corresponds to maximizing negative Kl divergence between $q^{t+1}$ and $p^\theta$. This then leads to the weighted negative log-likelihood objective. We have added this additional step in the paper too.

---

### Review · Reviewer_Bd9R · 2024-02-11

**Summary Of Contributions:**

The paper explores going beyond human data/feedback to improve language models on tasks such as math or coding problems where the correctness of the answers can be easily verified. Specifically, the authors investigate and propose a simplified version of an iterative self-training method by Gulcere et al. (2023) that involves generating samples filtering them based on binary feedback, and finetuning the model on them.  The main contribution is comparing PaLM 2 models trained with this method or with only human feedback in two domains, competitional-level mathematical problem-solving and code generation, as well as providing positive empirical findings for different model sizes.

**Audience:**

Yes

**Claims And Evidence:**

Yes

**Requested Changes:**

A few suggestions for strengthening the work
- The way this paper is positioned compared to ReST is somewhat confusing. In the introduction, it is implied that ReST is used as is but the model section introduces a few modifications with unexplored impact/consequences compared to the original version. It'd be good to provide a better motivation for the proposed modifications and surface them in the introduction.
- I'd suggest quantifying the human effort involved in writing the prompts with the effort required to provide human feedback to make the benefit more clear to the reader.
- Given that the ReST method is very closely related, it'd be useful to show in an ablation how the modifications made impact the performance by comparing to it.

**Strengths And Weaknesses:**

Strengths
- The examined self-training method is quite simple and improves language models of different sizes compared to relying only on human feedback.
- Introduces two simplifications to the self-training method by Gulcere et al. (2023)  by removing the reliance on human-generated outputs during the E-step and always finetuning the base model to avoid overfitting.
- Self-training leads to consistent improvements over models that have been trained only based on human feedback. The results hold for models of different sizes and models finetuned with this method lead to improvements on held-out benchmarks.

Weaknesses
- Even though there is reduced reliance on human feedback, the proposed method still requires numerous human prompts and a manual function to verify correctness. The main implication of this is that for new tasks there is significant effort involved and the process may not be easy to scale.
- Improvements from self-training saturate with a few iterations and sometimes lead to overfitting; this can potentially hinder its adoption by the community.

---

> ### Author Response · Authors · 2024-02-23
> **Response to reviewer**
>
> “The way this paper is positioned compared to ReST is somewhat confusing”
> - We have updated the introduction to highlight the fact that while our method shares similarities with ReST, there are some differences. We detail these changes through the addition of a new paragraph in Section 3. We first derive ReST$^{EM}$ by starting from the EMRL algorithm of Hinton and Dayan (1997), and then specify how this final derived algorithm differs from ReST.
>
> "It'd be useful to show in an ablation how the modifications (from ReST) impact the performance"
> - In Figure 7, we have added an experiment that highlights the downstream impact of the main modification from ReST. Overall, while ReST and ReST$^{EM}$ result in similar final performance on the APPS task (with ReST$^{EM}$ being slightly better), the transfer performance to HumanEval is substantially better with ReST$^{EM}$. Finally, we have also added a table in Section 4 highlighting the various algorithmic and experimental differences between our work and related works (ReST, STaR, RFT).
>
> “Quantifying the human effort involved in writing the prompts”
> - We use a simple 4-shot prompt for our experiments. The examples used in the prompt are taken from the training set of Hendrycks MATH and APPS datasets. While performing SFT requires thousands of human-written examples, the prompt only requires a handful of human written examples, so the effort needed to construct the prompt is about three orders of magnitude less than that of using human-written solutions for fine-tuning.
>
> “The proposed method still requires numerous human prompts and a manual function to verify correctness”
> - The correctness function is indeed a bottleneck toward applying ReST$^{EM}$ to more tasks. However, a substantial number of tasks do already come up with a correctness function (which is used for evaluating performance on these tasks), and at least for these tasks, we do not need additional human effort in designing a reward function. We agree with the reviewer that requiring a large number of human prompts is still a bottleneck, and have noted this as one of the major limitations of the work in the final section of the paper.

---

> ### Comment · Reviewer_Bd9R · 2024-02-27
> **Response to authors**
>
> Thank you for the response and the additional efforts.
>
> The methodological and empirical differences with REST are much more clear now. Regarding the effort involved, I appreciate
>  the cost assessment in the reply and I think it would be useful mentioning the extent of it somewhere in the paper.
>
> Other than that, my concerns have been addressed and I am pleased with the improved version.

---

> > ### Author Response · Authors · 2024-02-28
> > **Added note about costs.**
> >
> > We updated the paper with a note about the cost of experiments in Section 5.

---

### Review · Reviewer_sFSn · 2024-02-14

**Summary Of Contributions:**

This work showed the effectiveness and efficiency of training on model-generated data combined with a reward function for improving the performance of LLMs on problem-solving tasks such as math and coding. The authors also show the connection between Reinforced Self Training and expectation-maximization.

**Audience:**

Yes

**Broader Impact Concerns:**

I think for reasoning tasks it’s relatively safe to self train on model generated data. However for tasks that involve human opinions and preferences, it is unclear whether/how the proposed method would reinforce existing bias in the model. But I’m not sure whether that’s within the scope of this work.

**Claims And Evidence:**

Yes

**Requested Changes:**

I think adding error bars to as much as possible results would make the paper stronger, especially to Figure 8, and improve my willingness to recommend for acceptance.

**Strengths And Weaknesses:**

I am not an expert in this field and haven’t had any prior experience working in this field, so my evaluations are an educated guess.

Strengths:

Quality and significance: The authors conducted extensive experiments to show the effectiveness and efficiency of the proposed method, and indeed the performance is significantly improved with relatively low computational cost.

Originality: Assuming similar phenomena hasn’t been systematically investigated in prior works, the empirical results in this paper are novel. However, I find the idea to be a bit obvious, but I think this is made up by the extensive experimental results and ablation studies.


Weakness:
-	There’s no confidence intervals in all experimental results. For most experiments, the difference is big enough that it’s probably statistically significant. However for results like Figure 8, there are a lot of close results, and error bars there would provide more insights on what’s significant and what’s not.
-	The idea of finetuning on model-generated data is a bit obvious.

---

> ### Author Response · Authors · 2024-02-23
> **Response to reviewer**
>
> “I think adding error bars to as much as possible results would make the paper stronger, especially to Figure 8”
>
> - We agree error bars would be good to add, however running each fine tuning experiment costs on the order of thousands of TPU hours and so it would be prohibitively expensive to run enough seeds to get confidence intervals.
> - However, since Figure 8 is an external evaluation, we re-ran these evaluations using multiple checkpoints from ReST$^{EM}$ for both MATH and APPS fine-tuning runs, and have updated Figure 8 (now Figure 9) with the standard deviation computed from these evaluations. Our conclusions remain unchanged.

---

### Public Comment · ~Lili_Mou1 · 2025-07-12

Hello,

I am not an expert in machine learning, but can anyone tell whether this paper is a special case of REINFORCE?

The authors formulate RL with an equation below Eqn (1), which appears not to involve log p(*).  However, if one computes the policy gradient, it will have the same form as the gradient for Eqn (3), which is the proposed EM-RL.

So what's the difference between this paper and traditional REINFORCE (assuming the return is a final-step reward)?

Thank you.

---

### Decision · Action_Editor_SLvn · 2024-03-30

**Recommendation:** Accept as is

**Comment:**

The paper proposes a simple method called ReST𝐸𝑀, which generates samples from a language model and fine-tunes it using binary feedback. The paper evaluates ReST𝐸𝑀 on advanced MATH reasoning and APPS coding tasks using PaLM-2 models, and shows that it outperforms fine-tuning only on human data. The paper demonstrates that ReST𝐸𝑀 scales favorably with model size and improves generalization to other math and coding datasets.

All reviewers have given positive ratings. I recommend acceptance overall.

**Audience:**

Yes

**Claims And Evidence:**

Yes